# On the Criticality of Adaptive Boolean Network Robots

**DOI:** 10.3390/e24101368

**Published:** 2022-09-27

**Authors:** Michele Braccini, Andrea Roli, Edoardo Barbieri, Stuart A. Kauffman

**Affiliations:** 1Department of Computer Science and Engineering, Università di Bologna, Campus of Cesena, I-47521 Cesena, Italy; 2European Centre for Living Technology, I-30123 Venezia, Italy; 3Institute for Systems Biology, Seattle, WA 98109, USA

**Keywords:** Boolean networks, dynamical criticality, robots, adaptation

## Abstract

Systems poised at a dynamical critical regime, between order and disorder, have been shown capable of exhibiting complex dynamics that balance robustness to external perturbations and rich repertoires of responses to inputs. This property has been exploited in artificial network classifiers, and preliminary results have also been attained in the context of robots controlled by Boolean networks. In this work, we investigate the role of dynamical criticality in robots undergoing online adaptation, i.e., robots that adapt some of their internal parameters to improve a performance metric over time during their activity. We study the behavior of robots controlled by random Boolean networks, which are either adapted in their coupling with robot sensors and actuators or in their structure or both. We observe that robots controlled by critical random Boolean networks have higher average and maximum performance than that of robots controlled by ordered and disordered nets. Notably, in general, adaptation by change of couplings produces robots with slightly higher performance than those adapted by changing their structure. Moreover, we observe that when adapted in their structure, ordered networks tend to move to the critical dynamical regime. These results provide further support to the conjecture that critical regimes favor adaptation and indicate the advantage of calibrating robot control systems at dynamical critical states.

## 1. Introduction

Dynamical criticality has been identified as a feature of complex natural as well as artificial systems in plenty of settings [1,2,3], from cells [4,5,6,7] to artificial neural networks [8,9,10]. The main advantages that have been identified in systems working at critical regimes are the capability of achieving an optimal balance between (i) robustness against mutation and phenotypic innovation and (ii) their repertoire of behaviors and the reliability of their actions [3,11]. This capability is a desirable feature also in artificial adaptive systems that have to carry out a task in an unknown environment.

In this work, we focus on the role that criticality plays in robots undergoing an adaptation process. The robots adapt their controller—namely, a Boolean network—so as to improve their performance at a given task over time. This experimental setting makes it possible to investigate to what extent and in what conditions controllers based on critical Boolean networks have an advantage over networks in other dynamical regimes. Moreover, the specific adaptive mechanism used may shed light on the relation between criticality and phenotypic plasticity—a property characterizing organisms that, from one genotype, can express different phenotypes in response to different environments without involving genetic modifications [12,13].

Since their inception as an abstract model of gene regulatory networks [14], Boolean networks (BNs) have been the subject of a wealth of works investigating their computational and dynamical properties. Notably, BNs have demonstrated their ability to effectively capture significant biological phenomena, such as cell differentiation [15,16,17,18,19,20,21]. Evidence of an edge provided by critical Boolean networks has been demonstrated in classification, filtering and control tasks, just to mention some examples. BNs artificially evolved to solve combinatorial tasks of varying complexity have been shown to be driven toward criticality [22]. Benedettini et al. [23] studied learning BNs in different problems and contexts and found that criticality is important but not always necessarily needed, as the choice of the objective function and the idiosyncrasies of the problems play a major role in the attainment of a high performance. A recent work shows that critical random BNs used in signal processing make it possible to achieve more flexible and accurate results [24]. This latter work is of main interest to our application, as it uses BNs in *reservoir computing* systems, which are computational devices composed of a core (the *reservoir*) and an input and an output component. The reservoir is usually a non-linear dynamical system, such as a random recurrent neural network, while the readout layer is a simple linear network. The input signal perturbs the dynamics of the reservoir, which is translated into an output by the readout component. This latter is subject to training in order to perform classification or regression tasks. The peculiarity of these computational systems is that they exploit the inherent random dynamics of the reservoir, which then has to be sufficiently rich but robust so as to react to the external inputs in a reliable yet informative way to provide meaningful signals to the readout layer.

In this work, we build upon a previous preliminary investigation [25], where random BNs (hereinafter referred to as RBNs), picked in ordered, critical and disordered regimes, are used to control robots subject to an adaptive process so as to be able to explore an arena without colliding against walls and obstacles. The coupling between robot proximity sensors and a BN controller undergoes a simple adaptive procedure that retains only those combinations that improve the performance of the robot. Nothing else is changed during the adaptation, so the structure of the BNs does not change over time. The outcome of this study is that RBNs picked from the critical ensemble provide a significant advantage in the performance over ordered and chaotic (i.e., disordered) ones. This result is in accordance with previous results in reservoir computing [24], where the internal dynamics of critical RBNs was shown to be advantageous. Several questions are still to be addressed: Are there tasks in which the advantage of critical BNs as controllers is particularly marked? How does the adaptive mechanism that operates on the coupling between sensors (resp. actuators) of the robot and inputs (resp. outputs) of the BN compare to a usual mutational adaptive mechanism? Is there a tendency to criticality when BNs are subject to structural mutation?

In this contribution, we address these questions and try to identify general conditions that make critical BNs preferable as controllers in artificial agents and robots.

## 2. Materials and Methods

In this section, we detail the model used as robot control program, i.e., Boolean networks, along with the adaptive mechanisms and specific tasks introduced with the aim of addressing the questions posed at the end of the previous section.

### 2.1. Robots Configuration

This section describes the hardware and software structure of the robots.

*Robot Hardware Configuration*—We chose to use the foot-bot robot model [26], since it represents a simple but well-equipped robot. For our purposes, its relevant features are differential steering through two-wheel drive motor control; the presence of up to 24 proximity sensors that allow the detection of nearby objects; and up to 24 light sensors placed around the body of the robot that allow it to capture the intensity of light signals.

*Robot Control Software*—As in the Boolean network robotics (BN-robotics) approach [27,28,29,30], each robot uses a Boolean network as a software control program. In this way, the behavior expressed by it is a function of the internal dynamics of the network, which in turn is highly influenced by how it is connected to sensors and actuators.

In the following, we will refer to *input nodes* as the subset of nodes in the Boolean network used to perturb the state of the network with signals coming from the robot’s sensors. After an encoding phase, the state of these nodes will be overwritten with the values obtained from the sensors.

Instead, the *output nodes* will be those nodes of the Boolean network whose values will be used to control the actuators of the robot; each node will control a particular actuator.

So, at each simulation step, a robot (i) takes the values read from its sensors; (ii) encodes them into Boolean values using thresholds that depend on the nature of the sensor (see Appendix A for a detailed list of the parameters used in the experiments); (iii) uses these values to perturb, by overwriting, the input nodes of the Boolean network; (iv) performs a synchronous update of the BN; and (v) uses the values of the output nodes thus determined to control the actuators.

### 2.2. Experimental Settings

In this section, we present the tasks employed to address the previously mentioned research questions along with the details of the experimental settings.

The BNs used as control software in these experiments have n=1000 nodes, k=3 inputs per node and Boolean functions randomly generated by means of the bias parameter *p*. In particular, *p* will take values in {0.1,0.21,0.5,0.79,0.9} in order to investigate the difference in performance caused by control software operating, respectively, in ordered, chaotic and critical dynamic regions. Indeed, according to [31], ensembles of RBNs having k=3 are statistically ordered when bias is equal to 0.1 or 0.9, statistically chaotic when p=0.5 and statistically critical when bias is equal to 0.21 or 0.79. Only the BN nodes controlling the wheels have the function randomly chosen always with bias 0.5; this is to avoid naively conditioning the behavior of the robot, which would tend to be always moving (resp. resting) for high biases (resp. low biases). The tasks will be run 50 times each, each time with a different seed assigned to the pseudo-random number generator, in order to obtain statistically comparable results. A random seed affects the initial positioning of robots, the generation of Boolean networks and their initial states, and also the actions performed by the adaptation techniques. In each run, a population of 10 robots will try to adapt with only the computational capabilities at its disposal and without resorting to information sharing as in evolutionary computation.

The tasks to be accomplished are expressed through objective functions that the robots use during their adaptation phase. Each simulation is divided into 500 epochs with the duration of d=80 s (which corresponds to 800 simulation steps, since robots execute 10 steps per second). All the parameters used for the experiments are shown in Appendix A. At the end of each epoch, a robot determines the quality of its configuration on the basis of the score returned by the objective function. The score is accumulated for each simulation step and reset at the beginning of each new epoch. Each robot tries to maximize the score obtained in the various epochs by applying the predetermined form of adaptation. To do this, at the end of each epoch, the current configuration is maintained only if it is not worse than the previous one; therefore, *sideways moves* in the search space of Boolean networks are allowed.

***Task I***—The first task is a simple navigation with collision avoidance, and it consists of moving as straight as possible, minimizing the number of curves, and at the same time avoiding obstacles that may be encountered. The arena configuration employed for this task is represented in Figure 1. To obtain robots performing this task, we introduced the following objective function, which is typically used in evolutionary robotics [32]:(1)$$\frac{100}{E}\sum_{n=1}^{E}\left(1-\theta \left(n\right)\right)\cdot \left(1-\sqrt{|l(n)-r(n)|}\right)\cdot \frac{l(n)+r(n)}{2}$$

To allow robots to perceive obstacles, the 24 proximity sensors made available by the *foot-bot* robot model were logically aggregated into groups of size 3, keeping the maximum value for each group. Then, a threshold equal to 0.1 was used for each of the resulting 8 max values to determine the corresponding Boolean values, which will be equal to 1 if they are greater than the threshold and 0 vice versa.

***Task II***—The second task differs from the first task in the presence of two virtual regions that subdivide the arena, as depicted in Figure 2. Each robot will be assigned to a region in which to navigate and avoid obstacles, avoiding entering the wrong one. The score is calculated as in Equation (Equation 1) by introducing a penalty factor ϵ which takes the value 1 when the robot is in the correct region and −1 otherwise. The resulting objective function is the following:(2)$$\frac{100}{E}\sum_{n=1}^{E}\epsilon \left(n\right)\cdot \left(1-\theta \left(n\right)\right)\cdot \left(1-\sqrt{|l(n)-r(n)|}\right)\cdot \frac{l(n)+r(n)}{2}$$
The correct region is the one where the robot is initially placed during the deployment phase. In addition to the 8 proximity sensors previously introduced for Task I, a virtual sensor designed to perturb the dynamics of the Boolean network with the feedback on the region where a robot is currently located has been added.

***Task III***—It is a foraging task where robots must pick up a virtual object in the blue zone and bring it to the green zone while avoiding obstacles; see Figure 3. To do this, the robot is provided with a virtual hook controlled by a binary signal: *pick up* or *deposit*.

In addition, to help the robots orient themselves, a light source was placed over the green area to provide a gradient to follow. All this information is propagated over the Boolean networks through additional input nodes. The information of the region where a robot is located is encoded by the following Boolean signals: **00** for the neutral zone, **01** for the green zone and **10** for the blue zone.

Light signals, such as proximity signals, are propagated through 8 Boolean signals. In total, the robot network is perturbed with 8+8+2=18 Boolean input values (proximity, brightness, region), and 2+1=3 Boolean output values are used to control the actuators (motors, hook). The same objective function as in the previous tasks is used to push robots to move and explore the arena with the difference of further penalizing colliding robots (θ(n) multiplied by 2).

A reward *r* is introduced that is added to the fitness score whenever the robot performs one of these actions: (3)$$r=\left\{\begin{array}{cc} +50, & \mathrm{if}\;\mathrm{pick}\;\mathrm{up}\;\mathrm{while}\;\mathrm{the}\;\mathrm{robot}\;\mathrm{is}\;\mathrm{in}\;\mathrm{the}\;\mathrm{blue}\;\mathrm{region}\\ +50, & \mathrm{if}\;\mathrm{deposit}\;\mathrm{while}\;\mathrm{the}\;\mathrm{robot}\;\mathrm{is}\;\mathrm{in}\;\mathrm{the}\;\mathrm{green}\;\mathrm{region}\\ -100, & \mathrm{if}\;\mathrm{deposit}\;\mathrm{in}\;\mathrm{other}\;\mathrm{cases} \end{array}\right.$$
So, the resulting objective function for this task is as follows:(4)$$\frac{100}{E}\sum_{n=1}^{E}r\left(n\right)+\left(1-2\theta \left(n\right)\right)\cdot \left(1-\sqrt{|l(n)-r(n)|}\right)\cdot \frac{l(n)+r(n)}{2}$$
The *pick up* action is considered valid only if the robot is in the correct region, while the *deposit* action is only possible if the robot has previously collected a virtual object. In this task, robots do not have information about the object possession; they have to keep in memory the state they are in. The position of the hook may represent a feedback to exploit; indeed, if it is closed, the robot may assume to possess an object, but this is not guaranteed. This task, and the previous one in a similar but different way, is designed to force the control software to maintain a memory of this information and act accordingly. The need for memory is a design requirement that rules out the possibility of attaining simple, Braitenberg-like [33], robot controllers. Finally, note that in this task, it was necessary to further penalize colliding robots; for this purpose, we introduced a factor of 2 that multiplies θ.

For information on the interpretation of the parameters used in Equations (Equation 1), (Equation 2) and (Equation 4), refer to Section A.2 and especially Table A2. Meanwhile, a summary of the parameters used for the experiments carried out is shown in Table 1.

### 2.3. Adaptive Mechanisms

Here, we introduce the three adaptive mechanisms used: *in–out mapping*, *mutation*, and *hybrid*. Their differences are reviewed below:**In–Out** **Mapping**The internal structure of the Boolean network remains unchanged; what this mechanism changes is, at the end of each epoch, the coupling between BN nodes and robot actuators and sensors. In particular, driven by the objective function that we are going to present later, the robot replaces two input nodes and one output node in a random fashion. This causes the network to be perturbed by sensor values in different regions of the BN and the output for the actuators to be derived from a different combination of nodes.**Mutation** At the end of each epoch, the robot changes the internal structure of the Boolean network as follows: 1% of the truth table entries (80 bits, result of networks with 1000 nodes and Boolean functions with 8 entries each) is negated, and 1% of the connections between nodes (30 arcs) are redistributed. Both entries and edges are chosen randomly. The internal structure of the network thus changes, while the set of input and output nodes remains unchanged.**Hybrid** The combined effect of the previous techniques.

Note that both the hybrid and mutation techniques mutate the internal structure of the Boolean network.

## 3. Results

In this section, we will summarize the results of the comparison of adaptive mechanisms tested in the three presented tasks, focusing on the performance differences induced by robots controlled by Boolean networks taken from ordered, critical and chaotic ensembles.

### Comparison of the Adaptive Mechanisms

As stated before, we run 50 replicas for each configuration of parameters and collected the following statistics:**Mean of max objective function per** **epoch**To represent the robots’ ability to adapt as the length of the adaptation phase increases, we take for each epoch the maximum score (returned value by the objective function) achieved so far by the robot, taking into account all previous epochs. Subsequently, we calculate the average so as to obtain for each epoch, 500 in total, a point that represents the average of the maximum score obtained so far by all robots. The trend will always be increasing.**Area under the** **curve**To provide quantitative information about the differential performances of all the combinations of adaptive mechanisms and dynamical regimes, we calculate and rank from highest to lowest the area under the curve of the trend traced by the mean of the maximum objective function along the epochs.**Box-plot of max objective function** **scores**In this type of statistic, each box-plot is populated by the maximum objective function score of each robot. The maximum score is the highest score that a robot was able to reach in one of the 500 epochs available.**Mean rank of max** **scores**For this analysis, we sort all the robot’s max scores from highest to lowest and assign a rank. Afterwards, we compute the mean of the rank of every experimental case by grouping them by the mechanism and dynamical regime. For this statistic, the lower, the better.

In Figure 4, we present, for each task, the trend of the mean score per epoch, pointing out the relationship between the type of Boolean network and the adaptive mechanism.

First of all, we observe that Boolean networks generated with p=0.5 (blue ones) obtain, except for Task II, a lower average score regardless of the adaptive technique used. Secondly, we can also notice that the critical dynamical regime (green curves), in general, tends to dominate the others, with a slightly better performance of *in–out mapping* with respect to other adaptation schemas. To better characterize the contributions made, respectively, by the adaptive mechanism and the dynamical regime of the Boolean network in the critical controller case, we have to note the relative difference in performance obtained by the mutation mechanism compared to the other two. Indeed, the latter produces lower values than the other two mechanisms at each epoch, leading us to conclude that the results of the hybrid mechanism in the critical case were more positively affected by the change in input and output mapping than by the component concerning the Boolean network modifications. In contrast, the configuration of (initially) ordered controller (red curves) with hybrid technique appears to take advantage of the mutational component of the hybrid mechanism, except in Task I, which is the simplest one. However, how much these variations affect the dynamical regime of these networks is unclear, and in the following, we will analyze this aspect by analyzing their Derrida values.

As previously stated, to quantitatively evaluate the differences in performance, we also calculated the area under each curve shown in Figure 4. We then ranked them and reported the 5 best for each task in Table 2. Interestingly, the first 4 positions are occupied by critical RBNs, for all tasks, and on only one occasion does the mechanism of mutation appear. Furthermore, we note that in 2/3 of the tasks, the *in–out mapping* turns out to be the best for this type of statistic.

The overall picture seems to point toward the initial postulated hypothesis that the critical RBNs already have all the computational characteristics to perform the tasks, and therefore, the *in–out mapping* technique is sufficient to obtain good results.

In Figure 5, we report the distributions of the maximum objective function scores obtained by the robots along their epochs of adaptation, which are grouped by task. These data also confirm that Boolean networks generated with p=0.5 perform poorly on average. Indeed, the best robots in this parameter setup fail to achieve high values of the objective function across all the tasks. In addition, we can state that at least for this kind of analysis, there is no clear winner between the critical and ordered networks, which are the best configurations for the previous analysis. However, the box-plots seem to suggest that as the task difficulty increases, i.e., from *I* to *III*, the relationships between the two distributions gradually change in favor of the critical configuration. As regards the adaptation mechanism, both cases of critical RBNs (namely, p=0.21 and p=0.79) with *in–out mapping* present median values comparable to (in Task I) or higher to (Task II and III) other adaptation mechanisms.

The unpaired Wilcoxon statistical hypothesis tests, reported in Appendix C in Figure A2, confirm that the medians of the distributions of p=0.21 and p=0.79
*in–out mapping* are significantly different from the other distributions’ medians in Tasks II and III, with some exceptions in Task I.

As briefly introduced above, another prominent analysis is that concerning the **nominal** dynamical regime, i.e., the one statistically possessed by the networks generated randomly at the beginning of the adaptation phase, against the **actual** one, i.e., the operating dynamical regime expressed by the BNs of the best robots. The **actual dynamical regime** is the result of the particular adaptive pressure exerted by the objective function on the environment–robot system. Indeed, having undergone modifications to the internal structure through the mutation and hybrid mechanisms, Boolean networks belonging to the p={0.1,0.9} configuration that have performance comparable to the critical ones, initially ordered, may have undergone a change to their dynamical regime of operation, which, arguably, could have led them to resemble, dynamically, those in the critical regime. In general, structural modifications to the Boolean networks introduced by the *mutation* and *hybrid* adaptive techniques no longer allow us to consider the networks as part of the random Boolean network ensemble from which they were initially sampled, since the bias selected by the adaptation techniques could have undergone substantial changes. So, a precise characterization of the dynamical regime of the networks that have been subjected to adaptation is needed. In this regard, we study their Derrida parameter [34] λ at one step.

The procedure for computing the Derrida value for a single state of a Boolean network is as follows: a copy of the state of the Boolean network is created, and the value of a randomly chosen variable is negated; then, a synchronous update is performed for both states, and the Hamming distance between the two resulting states is measured. Usually, this procedure is repeated several times, for different variables, to determine an average value. The result measures the average propagation level of a perturbation and is statistically greater than 1 in chaotic networks, and it is less than 1 in ordered and 1 in critical networks. Since in our experiments, Boolean networks are embedded in robot bodies, and thus regions of their state spaces are not equally accessible due to physical constraints and the control program they are actually implementing, we compute, for each robot, the Derrida value using the following procedure:We identify the epoch in which the robot obtained the best score;We extrapolate the state of the network for all the 800 steps composing the epoch;We calculate the Derrida value for each of the 800 states and on all the possible perturbations, n=100. Then, an average value of all states belonging to the robot is computed;Finally, the average value is used to generate a point on the scatter plot.

The value thus calculated represents the average propagation level of a perturbation only on the states in which the robot has transited during its best epoch. Then, these values are reported in Figure 6 with respect to their relative scores. For reasons of clarity, Figure 6 reports the comparison of initially ordered, critical and chaotic networks generated using the following bias values: p={0.1,0.21,0.5}. The other bias values taken into account, namely p={0.9,0.79,0.5}, which show a similar trend, are reported in Appendix B and in particular in Figure A1.

In the cases where only an *in–out mapping* adaptive mechanism is applied, it is possible to notice well-divided clusters in relation to the Derrida value and to the parameter *p* used to generate the RBNs. When the mutation is introduced (*hybrid* and *mutation* techniques), the regions start to overlap. In particular, it is possible to notice that the green and red regions (p=0.21 and p=0.1, and in the same way for p=0.79 and p=0.9 in Figure A1) move to the right, obtaining a higher average Derrida value. Remarkably, the effect is particularly marked for the initially ordered networks. To better evaluate the effect of the “dynamical regime shift” observed in Figure A1) and induced by the mechanisms that change the BNs structure, we perform a further analysis: we compute the mean of the ranking of the maximum scores in both *nominal* and *actual* dynamical regimes. To do so, for the **nominal** dynamical regime, we considered as ordered the BNs generated with p={0.1,0.9}, critical those with p={0.21,0.79} and chaotic with p={0.5}. Meanwhile, for the **actual** dynamical regime, we considered them ordered if D<1−ϵ, critical if D≥1−ϵ and D≤1+ϵ, and, lastly, chaotic if D>1+ϵ, with ϵ=0.3. The results are reported in Table 3 for Task III, while those concerning the dynamical regime shift observed in Tasks I and II are presented in Appendix D, in Table A6 and Table A7. In general, we can note how all the configurations with (initially) ordered networks that are present in the first positions of the “nominal” column (remember that in this analysis, the lower, the better) are superseded by the critical ones when we consider the actual operating dynamical regimes expressed by the BNs that control the robots.

Overall, this result demonstrates how initially ordered networks that undergo adaptation by the hybrid and mutation mechanisms have statistically similar dynamical characteristics to the critical ones and strengthen our initial hypothesis.

## 4. Discussion

The results presented in the previous section can be considered as a further piece of evidence supporting the so-called *criticality hypothesis* [2,3,35,36]. In the following subsection, we first elaborate on that. Subsequently, we discuss the role of criticality in the broader context of phenotypic plasticity.

### 4.1. Remarks on the Criticality of Adaptive BN Robots

The results presented in the previous section are coherent with the hypothesis that criticality brings an advantage for the online adaptation of BN robots. As previously recalled, this advantage is commonly believed to manifest itself in two ways:(i)Behavior: optimal balance between the system’s repertoire of actions and their reliability with respect to external inputs.(ii)Evolvability: trade-off between robustness against mutation and phenotypic innovation.

Our experiments make it possible to set these properties in a more precise context, as they enable us to provide a formal instantiation of the objects involved. Let us first focus on “Behavior”. In our experimental setting, the BN is coupled with the environment via (a) an encoding layer that imposes temporary values on network nodes from sensors readings (the “external inputs”) and (b) a decoding layer that reads the values of some nodes and translates them into actuation signals (the “actions”). The actions the robot can take are low level, as they consist of activating the actuators (e.g., the wheels) for a short time duration, i.e., the interval between two simulation steps (in our experiments, 100 milliseconds). The sequence of these low-level actions produces the actual high-level actions, such as “going straight”. Therefore, the overall robot behavior is the result of the integration in time of low-level actions, which depend upon the inputs and the internal state of the network. Note that inputs may change both by robot movements (e.g., moving toward a wall makes it detectable by proximity sensors) and uncontrolled external events (e.g., other robots moving in the arena). The behavior of the whole system—i.e., the robot controlled by the BN—is quantitatively evaluated by means of a task-dependent objective function: The higher the evaluation, the greater the robot’s capability of elaborating inputs to produce useful high-level actions. In summary, property (i) can be defined as the capability of the BN—which is the robot control system—to discriminate useful inputs from useless ones and to translate them into proper time-extended signals to the actuators in such a way that a high performance with respect to the objective function is achieved.

As for property (ii), in our experiments, “robustness against mutation” has to be read as the aptitude of the network to react with slight behavioral changes to local permanent perturbations of its structure. In other words, on average, the dynamics of the network should not be profoundly changed by local structural perturbations. “Phenotypic innovation” counterbalances robustness in that it should enable changes that carry novelty in the dynamics of the network. In our experimental setting, this corresponds to the possibility of introducing adjustments to the network such that it can improve its behavior. Therefore, the trade-off between robustness against mutation and phenotypic innovation is indirectly assessed by the objective function improvements along the adaptive epochs: The larger the area under the curve, the better the trade-off between robustness and innovation—under structural adaptation, i.e., the *Mutation* mechanism.

We remark that the *in-out mapping* adaptive scheme does not perturb the network structure, but it intervenes on the functioning of the network by changing its connections to sensors and actuators. This mechanism does not change the dynamical regime of the network if taken as an isolated system, as indeed, it is commonly used to evaluate the BN dynamical regime. However, different couplings with sensors and actuators give rise to different dynamics: The same BN robot might react with different behaviors to the same environmental stimuli depending on the input–output connections. Therefore, our observations on the dynamical regime concern the isolated network rather than the one of the compound open system network–environment.

In the following, we discuss our results in light of the previous considerations. We first observe that RBNs in the chaotic regime are rather not suitable for successfully controlling adaptive robots, while in general, ordered and critical RBNs have shown noteworthy capabilities in this respect. The probability of attaining a sufficiently good performance with an RBN picked from the chaotic ensemble is indeed negligible, independently of the adaptation mechanism. Interestingly, this outcome is in agreement with previous results in eukaryotic cells, which were not found in the chaotic regime [37].

This is a further example of the difficulty of evolving or adapting a chaotic RBN; moreover, it provides further evidence to the general conception that chaotic systems do not react reliably to external inputs. In our experiments, the latter sentence takes a precise meaning: a good performance is associated with the capability of capturing the regularities of the external world, filtering out noise and perturbations, and transforming them into actions coherent with the final goal. Chaotic RBNs do not provide such ability to the robots they control. Conversely, RBNs picked from the critical and ordered ensembles have shown to be able to control robots achieving high performance although with some remarkable differences. First and foremost is that critical RBNs achieve a slightly better overall average performance than ordered ones (Figure 4). However, if we look at the best solutions (Figure 5), or better at the mean of their ranks (Table 3), the results are clearer and lead us to conclude that critical RBNs achieve superior performance with respect to other regimes.

A further point to be remarked is that the best results achieved by robots controlled by critical RBNs are attained under the *in-out mapping* adaptive mechanism, indicating that the intrinsic dynamics of critical RBNs constitutes an abundant repertoire of actions, as in the case of reservoir computing [24]. This is a distinguished feature of those systems that are characterized by a rich dynamics but are robust against external perturbations. Conversely, ordered RBNs, when the difficulty of the task becomes not negligible (i.e., Task II and III), greatly benefit from the adaptive schemes that change the topology and Boolean functions (i.e., *Mutation*, and *Hybrid*, which encompasses the former). This result is further reinforced by observing that initially ordered RBNs tend to move toward the critical regime when their structure is evolved. Plots *g–i* of Figure 6 show the Derrida value distribution of RBNs that do not undergo structural changes; hence, they represent the initial distribution of Derrida values for RBNs in the three regimes. Roughly, ordered network values are centered on 0.5, critical on 1.0 and chaotic on 1.5. If we compare these plots with the ones of *Mutation* and *Hybrid* adaptive schemes, we observe that the cloud of points corresponding to ordered RBNs tends to move right, i.e., toward the critical regime. One of the reasons for this phenomenon is the drift toward Boolean function bias equal to 0.5 imposed by random mutations in the Boolean functions. In fact, we observe a similar tendency also in critical RBNs. However, in ordered RBNs, this effect is considerably more pronounced, suggesting that other causes come into play. The most likely candidate is the adaptive pressure toward higher objective function values. To prove this conjecture and thus better frame the relative effect of this “shift toward chaos” induced by random mutations with respect to the adaptation pressure exerted by the objective function, we formulate a simple but effective model of dynamical regime change along the epochs of adaptation. The aforementioned model together with the relative results are present in Appendix E. In particular, Figure A3 shows the results of applying this model with the parameters used in our experiments. Based on the results obtained, it is clear that a complete shift toward chaotic dynamical regimes is already observed starting from the epoch number 200 in the absence of adaptation by the objective function. This leads us to conclude that in our experiments, this is not the dominant force, but that adaptive pressure is instead effective in shaping robots’ behaviors.

A final consideration concerns the results achieved under the *Mutation* adaptive mechanism: in all the tasks, critical RBNs attain the best average performance. This supports the advantage of criticality in evolutionary settings (property (ii)).

### 4.2. Criticality and Phenotypic Plasticity

Our results show that the highest performance in adaptive robots controlled by critical RBNs is attained under the *in–out mapping* adaptive mechanism. This is a notable finding, as it demonstrates that the repertoire of dynamics in a critical RBN is rich and flexible enough to be successfully tailored to the task at hand without the need for changing either the Boolean functions or the connections between nodes, or both. This property can be rephrased in more abstract terms as the capability of the same system to produce different behaviors in the face of different environmental conditions and inputs. In biology, the capacity of the same genotype to produce different phenotypes depending on the environment where it lives is defined as *phenotypic plasticity* [12,13]. The discussion of this phenomenon is out of the scope of this contribution, but it is important to emphasize that from the cybernetics standpoint, the ability of adapting one genotype to produce different phenotypes resembles the capability of producing an internal model of the external world [38]. In abstract terms, this process makes it possible for an organism to compress the wealth of information coming from the external world into an internal representation that values only the pieces of information relevant to the organism’s survival. On the basis of this internal model, the organism acts so as to accomplish its missions, in primis to attain homeostasis, i.e., maintaining its *essential variables* within physiological ranges [38]. We can then state that phenotypic plasticity not only is a vital property for living organisms, but it may also be of great value for artificial ones that need to be autonomous.

The *in–out mapping* adaptive mechanism we have implemented can be said to be a case of phenotypic plasticity. The main reason is that in our setting, adaptation involves only the way external information is filtered by the robot; hence, it concerns the sensing module of the system. This adaptation takes place during the “life” of the individual, and it is based on a feedback that rewards specific behaviors (i.e., those rewarded with high objective function values) without changing the actual “genetic” structure. This mechanism mimics a kind of sensory development tailored to the specific environment [39,40]. In general, the robot can be coupled with the environment in a huge number of possible combinations, each constraining the system to express a particular behavior (i.e., phenotype). The mapping between sensor readings and network nodes is the result of the embodied adaptation of the sensory-motor loop and manifests one particular phenotype, emerging from the interaction between the robot and the environment.

In addition to providing evidence to the criticality hypothesis, the results we have presented make it possible to speculate further: criticality might be an advantage because it enables phenotypic plasticity. We believe that the outcome of our experiments provides a motivation for a deeper investigation. We also envisage the possibility of devising wet lab experiments, in which the dynamical regime of an organism is externally controlled and its ability to exhibit phenotypic plasticity can be estimated.

It is important that it is now well established that real gene regulatory networks are, in fact, dynamically critical [6,7]. The cortex is also critical [41,42,43]. Thus, two major highly evolved systems have evolved to attain critical behavior. It seems reasonable to suggest that dynamical criticality will be of broad use in adaptive robots.

## 5. Conclusions

In this work, we have studied the performance of robots controlled by RBNs from ordered, critical and chaotic regimes, in three tasks of varying complexity. We have found evidence that the ensemble of critical RBNs (1) produces the best average results; (2) produces the best maximum results as well; (3) can be adapted without changing their internal structure; and finally, (4) shows the best average performance among the three regimes in the case of the *Mutation* adaptive mechanism.

In the frame of our experimental setting, these outcomes reinforce the hypotheses of the advantage of critical systems in accomplishing tasks in an environment. However, our findings are somehow indirect: the advantage of being critical is assessed by means of the performance quantitatively evaluated in terms of an objective function. The actual internal processes that led to these achievements still need to be studied and put in the context of information processing in critical systems [44]. A possibility is to investigate the relation between the criticality of controlling BNs, their performance and the maximization of some information theory measures, such as predictive information [45], integrated information [46] and transfer entropy [47].

Finally, in light of the great complexities posed by the path-dependent nature of the online adaptation process [20,48,49], we conclude that a mechanism such as the one we have introduced might be an effective tool for tuning artificial systems to the specific environment in which they have to operate. As a futuristic application, we imagine the construction of miniaturized robots that can accomplish missions precluded to humans, such as recovering polluted environments. In fact, recent technological advances have made it possible to build incredibly small robots to the size of tens of nanometers. The current smallest robots—built by biological matter—can perform only a few predetermined actions (see, e.g., the recent prominent case of Xenobots [50]); therefore, they cannot attain the level of adaptivity and robustness needed for a complex mission. On the other hand, Artificial Intelligence software has recently made tremendous advancements and has been proved capable of learning and accomplishing difficult tasks with a high degree of reliability. This software, however, cannot be run onto tiny robots. A viable way for filling this gap is provided by control programs based on unconventional computation, such as the ones derived from cell dynamics models, where phenotypic plasticity may play an important role.

## Figures and Tables

**Figure 1 entropy-24-01368-f001:**
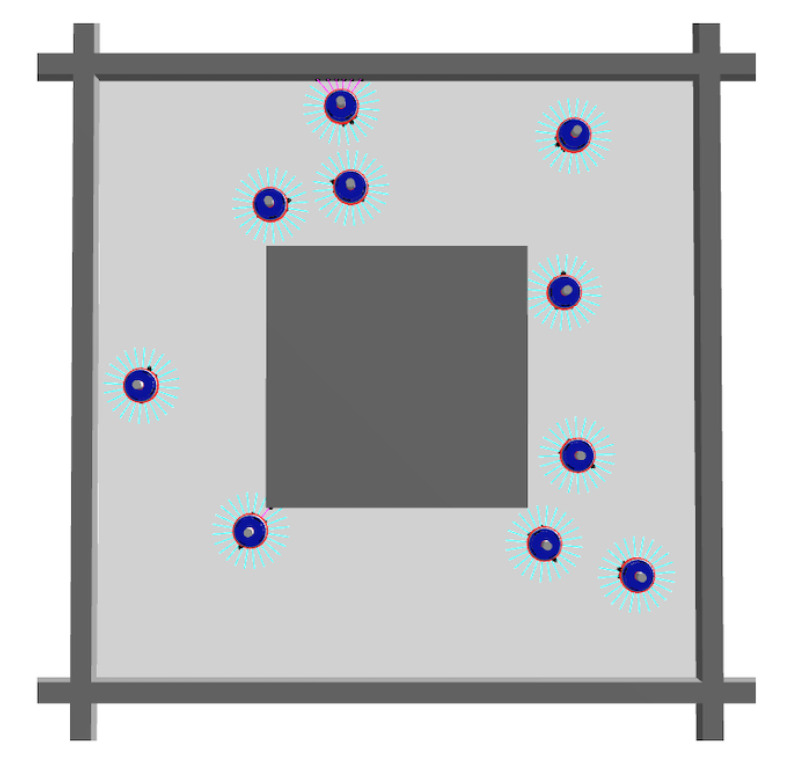
Top view of the arena used for Task I. The gray objects are the obstacles, and the blue objects are the robots. The arena consists of a square perimeter with a square obstacle in the center.

**Figure 2 entropy-24-01368-f002:**
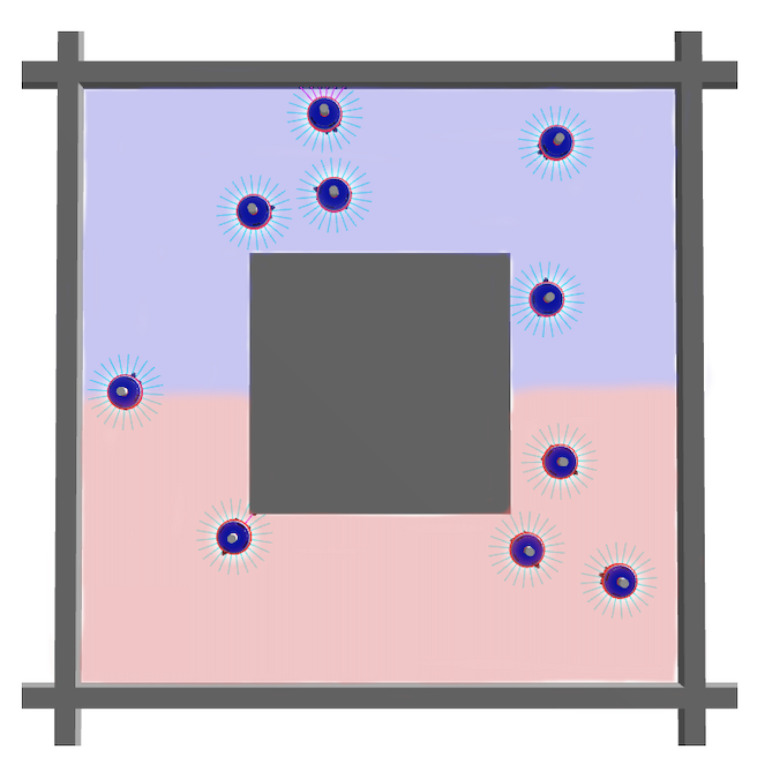
Top view of the arena used for Task II. As in the first arena, the gray objects are the obstacles, the blue objects are the robots and the arena consists of a rectangular perimeter with a square obstacle in the center. Unlike the first arena, this one has, in addition, two virtual areas, red and blue.

**Figure 3 entropy-24-01368-f003:**
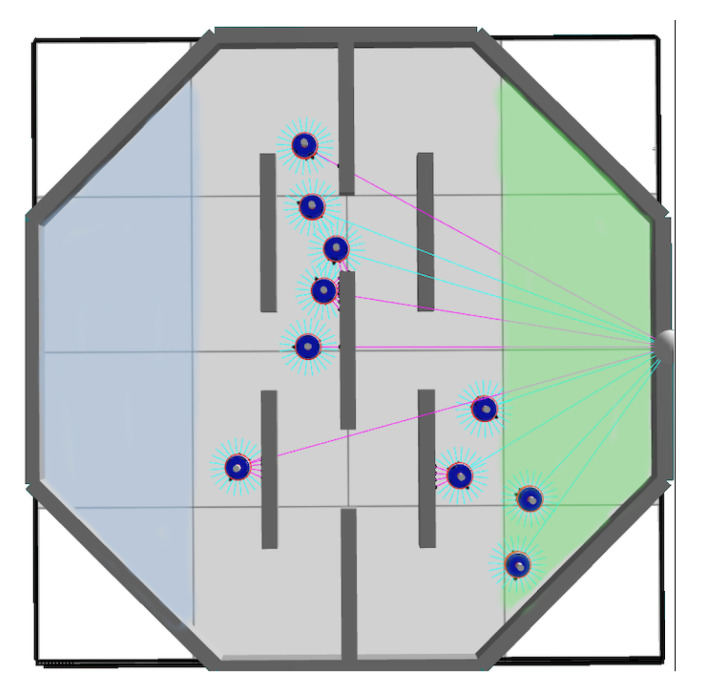
Top view of the arena for Task III. The third arena features an octagonal perimeter with obstacles in the center; the blue and green areas indicate two virtual areas needed to accomplish a foraging task. Moreover, it also presents a light source located on the outer boundary of the green area. Purple and green lines indicate whether or not the robots are perceiving light and were used for debugging purposes.

**Figure 4 entropy-24-01368-f004:**
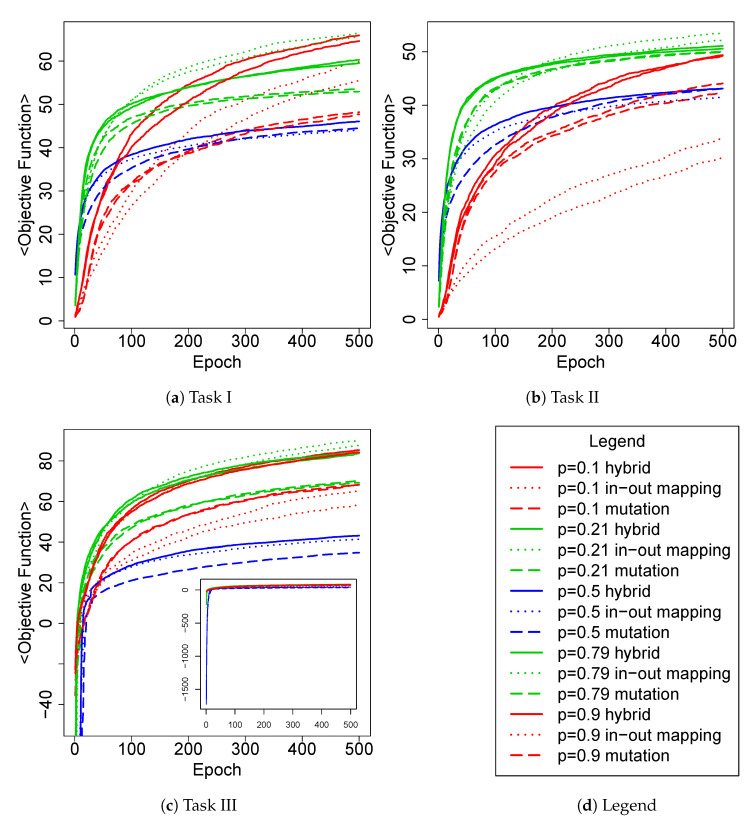
Comparison of average maximum score trends obtained in each epoch by all robots with respect to the *p* parameter and the three adaptive techniques. Graphs are divided by the task they address. Given the very low average max score obtained by chaotic parameters in the first epochs (subplot), the Task III main graph shows the values above to −50.

**Figure 5 entropy-24-01368-f005:**
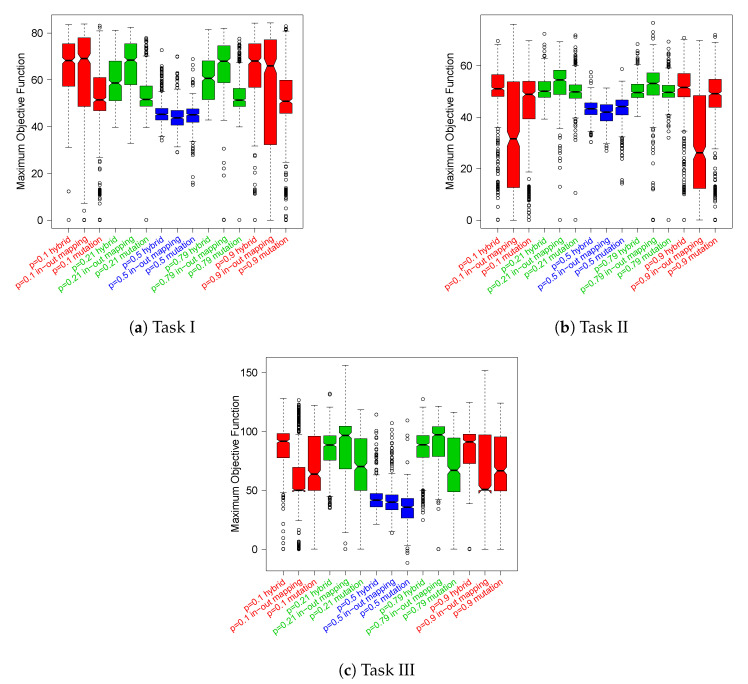
Comparison of the actual distributions of maximum objective function values obtained by all the robots, subdivided by task typology.

**Figure 6 entropy-24-01368-f006:**
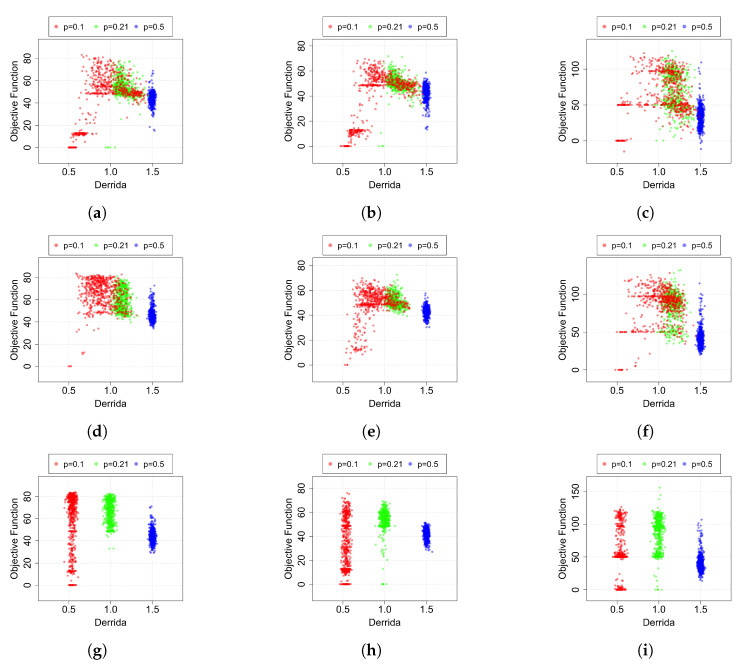
Scatter plot of Derrida values (x-axis) against the relative objective function scores (y-axis) obtained by the robots in their best epoch. Noting that we used RBNs with k=3 as robot controllers, red points represent Boolean networks initially generated using p=0.1, while the green dots represent RBNs generated using p=0.21, and lastly, blue ones have been sampled from the ensemble of chaotic RBNs, i.e., using p=0.5. (**a**) **Task I**: *Mutation* (**b**) **Task II**: *Mutation* (**c**) **Task III**: *Mutation* (**d**) **Task I**: *Hybrid* (**e**) **Task II**: *Hybrid* (**f**) **Task III**: *Hybrid* (**g**) **Task I**: *In-Out Mapping* (**h**) **Task II**: *In-Out Mapping* (**i**) **Task III**: *In-Out Mapping*.

**Table 1 entropy-24-01368-t001:** Summary of the parameters used for the experiments.

n=100, k=3, p∈{0.1,0.21,0.5,0.79,0.9}, (no self loops allowed)
Δt=0.1, d=80 s;
Population of 10 robots per each run;
Adaptive mechanisms ∈{In-Out Mapping, Mutation, Hybrid};
Tasks ∈{I, II, III};
50 runs for each parameters combination

**Table 2 entropy-24-01368-t002:** Ranking, for each task, of the first 5 values of the area under the curves of the trends shown in Figure 4. [**i-o**] stands for *in–out mapping* mechanism, while [**h**] stands for hybrid adaptation and [**m**] stands for mutation scheme.

	Task I		Task II		Task III	
Rank	Case	Area	Case	Area	Case	Area
1	[**i-o**] p=0.21	28,431.61	[**h**] p=0.21	23,302.72	[**i-o**] p=0.79	35,406.36
2	[**i-o**] p=0.79	27,921.29	[**i-o**] p=0.21	23,283.64	[**h**] p=0.79	34,519.52
3	[**h**] p=0.79	26,603.61	[**h**] p=0.79	23,148.49	[**i-o**] p=0.21	33,832.64
4	[**h**] p=0.21	26,414.22	[**m**] p=0.79	22,482.42	[**h**] p=0.21	33,781.08
5	[**h**] p=0.1	26,079.86	[**i-o**] p=0.79	22,410.69	[**h**] p=0.1	33,630.86

**Table 3 entropy-24-01368-t003:** Mean of the ranking of the maximum scores for Task III; in this analysis, the lower, the better. [**i-o**] stands for *in–out mapping* mechanism, while [**h**] stands for hybrid adaptation and [**m**] stands for mutation scheme. For the **nominal** dynamical regime, we considered as ordered the BNs generated with p={0.1,0.9}, critical those with p={0.21,0.79} and chaotic those with p={0.5}. Meanwhile, for the **actual** dynamical regime, we considered them ordered if D<1−ϵ, critical if D≥1−ϵ and D≤1+ϵ, and, lastly, chaotic if D>1+ϵ, with ϵ=0.3.

Task III				
	Nominal		Actual	
Rank	Case	<Rank>	Case	<Rank>
1	[**i-o**] *critical*	2593.62	[**i-o**] *critical*	2600.79
2	[**h**] *ordered*	2975.89	[**h**] *critical*	2956.95
3	[**h**] *critical*	3185.77	[**m**] *critical*	4002.53
4	[**m**] *critical*	4366.05	[**i-o**] *ordered*	4779.87
5	[**m**] *ordered*	4435.57	[**h**] *ordered*	5468.71
6	[**i-o**] *ordered*	4804.36	[**m**] *ordered*	6687.97
7	[**h**] *chaotic*	6967.89	[**h**] *chaotic*	6734.92
8	[**i-o**] *chaotic*	7134.18	[**m**] *chaotic*	7063.64
9	[**m**] *chaotic*	7557.92	[**i-o**] *chaotic*	7134.17

## Data Availability

Code and data of the experiments discussed in this paper are available at https://github.com/edobrb/progetto-irs (accessed on 1 August 2022).

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
