# Peer review of "On the Criticality of Adaptive Boolean Network Robots"

_entropy, 2022, doi:10.3390/e24101368_

Round 1
Reviewer 1 Report
Disclaimer: I am very familiar with Boolean networks but have not worked on adaptive robotics.
This manuscript describes a very interesting, well-conducted experiment. It is well-written and most of my confusion (see major comments) was resolved after I finished reading through the manuscript. It would help readers if the authors could add more references to the supplementary material in the text, as this would have prevented my confusion. Other than that, I have mainly stylistic comments. Some of the methods could also be better, more extensively explained. Lastly, the description of the Github repo exists only in Italian. This may constitute an issue for some readers. Below a few major comments, followed by minor comments and language suggestions.
Major comments:
Equation 1: This equation is key to understanding the action of the robots in Task 1, yet none of the parameters in this equation are further defined or explained. This needs to be added, probably in a separate paragraph, or if common knowledge in the field of robotics, requires at least proper citation.
Edit: After reading through the whole manuscript, Table A2 finally provides some help at understanding Equation 1. This table needs to (a) be referenced at the appropriate position in the main part, and (b) requires still further explanation. E.g., what is the effect of a left or right motor that takes on the value 1?
Through random ``neutral" mutation (flipping of bits), the bias of any robot should approach 0.5 in the long run. The benefit of operating in the critical regime may prevent this from happening. A mathematical analysis that compares how quickly on average the bias approaches 0.5 with the observed bias in top performing robots over the 500 epochs would be interesting.
Edit: Figure A3 shows exactly this. This figure needs to be referenced at the appropriate position in the main part. I also think the last two sentences (l530-534) belong in the main text.
Minor comments:
l100: the latter part of the sentence ``in the case of the two motors controlling the wheels, the signal 1 is interpreted as full power and 0 vice versa" seems out of place, or at least requires a little more explanation. Why do two motors need to be specified here? What about the case of four motors? To a non-robotics person, this is completely unmotivated here.
l106: this is the first mention of ``epoch". It should be explained here.
2.2. Mutation: For a while, I was confused by the numbers in this subsection. I think it is clearer to replace ``with 8 entries each" by ``with 3 inputs each", or maybe add both to really clarify where the total of 8000 bits and 3000 arcs come from.
Both the above mentioned concerns could be resolved simply by switching 2.2. and 2.3. I see no disadvantage to this suggestion.
Figure 1:
(a) The arena shown is a square. In the figure legend it is described as rectangular, which is certainly not wrong but may not be as precise as possible. (b) Several sentences in the figure legend do not refer to this figure. Delete.
(c) Third and fourth arena as well as last two arenas is confusing. E.g., what is the second arena?
l151/152: the aggregation described here requires a little more explanation. An example would be helpful, particularly one that highlights what exactly the sensors sense and how does this signal gets turned into a Boolean value. I have a guess but the text does not provide enough detail as is.
l171/172: the authors defined input and output values previously. It would be nice to use these terms here. E.g., ``perturbed with 8 + 8 + 2 = 18 Boolean INPUT values", ``2 + 1 = 3 Boolean OUTPUT values"
Figure 3: The meaning of the purple lines is not explained in the figure legend.
l235: I believe this sentence is not fully correct: ``Interestingly, all 5 top positions are occupied by critical RBNs, for all tasks". The fifth best mechanism for Task I and Task III has p=0.1, which corresponds to an ordered regime.
l286: This sentence requires a little more explanation, some more details on the extrapolation. Also, why 800 steps in this sentence and 800 states in the next? Is this one and the same?
Table 3 legend: delete the word Clearly, also in corresponding suppl. tables.
l505: this is the only mention of ARGoS. It needs to be spelled out and/or requires a citation.
Table A5 states that p (the bias?) of output nodes is 0.5. What does this mean? Why was this choice made? I couldn't find any explanation. Does this choice affect the results? If so, how?
The description of the code on github is provided only in Italian, not English.
Language / Typos:
l89: input nodeS should be plural
l134: this sentence is a little confusing. Why not write: ``In each run, a population of 10 robots will try to adapt with only the computational capabilities at its disposal, and without resorting to information sharing as in evolutionary computation."
l148: space between Figure and 1.
l237: replace cases by tasks
l255: distribution's medians
Figure legend A2: the first sentence is probably missing a word, e.g. ``Graphical representation of the results of the Wilcoxon test FOR the distributionS presented in Figure 5"
l287: replace ``value of Derrida" by ``Derrida value"
l342: actionS should be plural
l409: delete one also
l450: sensor readings
l483: robotS should be plural
l508/09: awkward sentence/grammar.
l526/27/28: there seem to be some words missing here. The sentence doesn't make sense.
Reviewer 2 Report
This paper studies the performance of robots controlled by random Boolean networks (RBNs). RBNs are coupled with the sensors and actuators of robots. The couplings or the structure of RBNs are adapted online for robots to have better performance. The results obtained from the three experiments suggest that robots controlled by dynamically critical RBNs have higher performance than those controlled by ordered or chaotic ones.
This is a new result supporting the criticality conjecture. It seems that the experiments are carefully designed and subsequent analysis is scientifically sound. Thus, the reviewer recommends acceptance of the paper. However, the following specific comments should be considered before publication.
=====Specific Comments=====
1) Sec.2.2 and 2.3:
It seems that how each robot tries to maximize the score is not described. Please explain the maximization algorithm for each adaptive mechanism.
2) Page 4, Eq.(1):
Please define the symbols in Eq.(1) and explain the meaning of each term. Although the symbols except `epsilon(n)' are summarized in Table A.2, it is more convenient for the readers to define and explain them also in the main text.
3) Page 4, Figure 1:
The fourth and subsequent sentences in the caption seem to have nothing to do with the figure. Please check them.
4) Page 16, Line 505:
The abbreviation `ARGoS' appears without explanation. Please explain it.
5) Page 19, Eq.(A1):
Please define `x'.
6) Page 19, Eq.(A1): The same symbol `E[N_{t+1}^1]' is used for two different things(the first and the last lines of the equation). Please fix this.
7) Page 21, Figure A3:
The equality in `N_{t=0}^1 = ...' in the second and third lines in the caption should be replaced by the set membership symbol.
